# The Effect of Stress on Individuals' Wasting Behavior: The Mediating Role of Impaired Self-Control

**Ke Zhang** [ID] **and Yuanyuan Cai** *[ID]

SILC Business School, Shanghai University, Shanghai 200444, China; zhangke1988swz@gmail.com
* Correspondence: cyy724@shu.edu.cn

**Abstract:** Wasting behavior has become a serious issue in modern society, especially when individuals face economic recessions and environmental problems. Despite the literature exploring cultural and sociological antecedents of wasting behavior, limited attention has been given to the role of individuals' associated psychological states. The present research fills this gap by examining how and why stress, a psychological state pervasive among people in the modern world, can influence individuals' wasting behavior through three studies. Pilot study and Study 1 provide evidence of the positive relationship between stress and wasting behavior. Then, Study 2 sheds light on the mechanism underlying the proposed effect by taking impaired self-control as a mediator. Lastly, the theoretical contributions and practical implications of this research are discussed.

**Keywords:** stress; wasting behavior; self-control; waste management; psychology





## 1. Introduction

Wasting behavior is defined as inefficient resource use, in accordance with prior work [1–3]. Arkes [1] posited that people can be considered wasteful when they discard or do not fully use previously purchased items. In particular, he provides an example of a person who purchases a doughnut but only eats half of it. Consistent with this viewpoint, Bolton and Alba [2] asserted that wastefulness occurs when a person possesses unused or leftover resources. Various scholars have developed targeted measures of wasting behavior based on this definition. For example, Visschers et al. [4] evaluated wasting behavior by asking participants to report whether they finished consuming food they had purchased. Lin and Chang [3] assessed wasting behavior by quantifying inefficient paper use when evaluating the quality of scissors. Additionally, Hamilton et al. [5] asked Australian residents to estimate their expenditure on purchased goods and services that were never or hardly used, representing an indicator of wasteful consumption. Therefore, such definitions and measures of wasting behavior center on the use of purchased or owned items, which involves post-purchase behavior and differs from pure purchase behavior such as impulsive consumption [6,7].

Although many people are averse to being wasteful in their consumption [2,8], tons of resources, such as food and water, are nevertheless wasted every year [3,9]. As estimated by United States Environmental Protection 2020 [10], 63.1 million tons of food waste were generated in the commercial, institutional, and residential sectors in 2018, accounting for 21.6% of the total municipal solid waste generation in the United States. In China, a national survey conducted by the National Peoples' Congress 2020 [11] revealed that the urban restaurant waste had exceeded 17 million tons.

Given the importance of resource conservation in today's crowded world (especially amid the salient threat of resource scarcity), many researchers have begun to investigate why people waste resources. This research area has revealed several major categories of factors that drive wasting behavior, including (1) sociodemographic attributes, such as household income [12]; (2) marketing stimuli, such as package design [13]; and (3) cultural aspects, such as domestic food practices and social norms [14].

Although the aforementioned literature analyzed wasting behavior from sociological and cultural perspectives, related work has seldom considered the roles of individuals' psychological states (e.g., emotions and feelings) in triggering such behavior [15]. For example, Porpino [16] recommended that research on wasting behavior should explore the influences of specific emotions, such as guilt and sadness, on humans' food wastage. To date, however, few articles have addressed this area. In the current research, we propose that stress, as a prevailing psychological factor in the modern world, will lead to more wasting behavior.

Such an exploration makes several major contributions. First, wasting behavior is an emerging problem in economics and public policy [16,17]. The present research offers a novel perspective on this subject by identifying stress (i.e., a negative psychological state) as an antecedent of individuals' wasting behavior. Second, an investigation of this topic can enhance our understanding of the behavioral consequences of stress. Third, our work sheds light on the mechanism underlying the proposed effect via mediation approach, which should enable policymakers to strategize to attenuate this effect. In this case, our findings also provide valuable practical implications in terms of waste reduction.

In the following parts of this paper, we will first go through the concepts and research about stress, and then discuss how and why stress can magnify people's wasting behavior. Specifically, we suggest that this reaction is due to the impaired self-control among people experiencing stress. Then, in the methodology part, we will present one pilot study together with two main studies, aiming to provide robust evidence for our proposed hypothesis.

## 2. Literature Review

### 2.1. Research about Stress

Stress is common in modern society. According to a nationwide study by the American Psychological Association (APA) in 2020 [18], more than 60% of surveyed Americans reported experiencing high stress. Scholars have defined stress as a psychological state experienced by people confronted with demanding, challenging, and threatening events or situations, also termed stressors [19,20]. Research has shown that stressors can be either incidental challenging events or chronic threatening situations [20,21]. For example, major life events such as college entrance examinations or divorce, and environmental factors including natural disasters or air pollution, serve as stressors that contribute to stressful feelings [21–23].

As a complement to studies on the antecedents of stress, prior work has also investigated its consequences. For example, some researchers have explored how stress can influence individuals' physical health [24,25] and mental health [21]. For instance, Mazure [26] reviewed the effects of life stressors on depression in humans, and Gradus et al. [27] later found that stress disorder was directly associated with suicidal behavior. Other studies have delineated the impact of stress on cognition. For example, stress has been shown to impair humans' cognitive flexibility [28], attention capacity [29], and memories [30]. Li et al. [20] discovered that stress could bias individuals' numerical perceptions and judgments. In addition, stress leads to behavioral inhibition. Specifically, previous studies have found that experiencing stress stimulates cortisol release, which can enhance avoidance motivation, manifesting in defensiveness and introversion in social life [31–33]. In the workplace, it also results in withdraw behaviors like job burnout [34] and turnover intention (Liu and Onwuegbuzie, 2012) [35]. Despite earlier research examining the influence of stress on humans' social behavior (e.g., social interaction) [33,36,37] and health-related behavior (e.g., drug addiction) [38,39], limited attention has been paid to the impact of stress on consumers' everyday behavior [15,19]. In fact, consumers make choices to satisfy their basic human needs. For example, a joint experience of fear and disgust increases consumers' preference for well-known or commonly used products, because these familiar products can provide them with a sense of certainty and control [40]. When consumers' social status is threatened, they have a strong desire for products that signal high status, such as luxury products [41,42]. Therefore, a deep understanding of the psycho-

logical process driving consumers' behavior can lead consumers to maximize their utility when making consumption decisions. Wasting behavior is associated with spending more than the required amount on products, which means that consumers do not maximize the utility of money [1]. In the present study, we expect to explore the psychological process by which wasting behavior occurs. In addition, diverging from those aforementioned research streams on stress, we focus on the behavioral consequences (i.e., wasting behavior) of stress. Specifically, we attempt to test the effect of stress on people's wasting behavior and to investigate the mediating role of impaired self-control.

*2.2. The Effect of Stress on Wasting Behavior: The Mediating Role of Impaired Self-Control*

Self-control refers to the voluntary process through which individuals overcome urges or temptations that impede their goal pursuit [43,44]. Self-control is associated with personal well-being [43,45] and is regarded as a hallmark virtue of humanity [46,47]. However, various stressors in daily life, such as natural disasters and financial constraints, can threaten humans' self-control in lab and real-world contexts [29,48,49]. For example, Bernheim et al. [50] argued that poverty, a pervasive type of stressor, was associated with low self-control. Baumeister et al. [51] noted that stressors in social relationships, such as social exclusion, can impair one's self-control. Oaten and Cheng [52] demonstrated that stress from academic examination would undermine students' performances in tasks requiring self-control. Thus, it is reasonable to presume that stress in general can threaten humans' self-control.

In the present research, we propose that impaired self-control induced by stress can magnify people's wasting behavior. Our proposition is built around a discussion about the defining characteristics of wasting behavior. By definition, wasting behavior refers to inefficient resource use [1]. Extensive research has shown that wasting behavior is associated with self-regulatory failure. For example, Vohs et al. [53] contended that one consequence of self-regulatory failure was the less appropriate use of (and thus wasting) resources such as time and money. Scholars have also found that sustainable behavior, such as lowering the volume of waste produced, is contingent on self-control [54]. In support of this viewpoint, scholars have pointed out that successful self-control requires voluntary efforts invested into planning and executing behavior [55,56]. These effort investments are also necessary if humans want to reduce wasting behavior [3,57]. Relevant literature has also provided support for this proposition by showing that self-reported wasting behavior is higher when people are less able to follow their own plans and regulate their own behaviors [4,57–60]. Based on the discussions above, we propose the following hypotheses:

**Hypothesis 1 (H1).** *High (vs. low) stress will lead to more wasting behavior.*

**Hypothesis 2 (H2).** *The proposed effect of stress on people's wasting behavior is mediated by impaired self-control.*

## 3. Method and Results

We tested these hypotheses by conducting three studies. First, a pilot study was used to explore the correlation between stress and wasting behavior. Then, Study 1 aimed to provide coherent evidence for H1 by using different samples and measures. In Study 2, we measured the level of stress and tested how it influenced people's wasting behavior with impaired self-control as the mediator, aiming to examine H2.

*3.1. Pilot Study*

3.1.1. Procedure

In this study, we measured stress experienced by employees at their workplace and predicted that it would be positively correlated with self-reported wasting behavior. We predetermined the sample size by following several steps. First, a priori power analysis using G*power software [61] showed that at least 82 participants would ensure a medium-sized

correlation (i.e., 0.3) between stress and wasting behavior, with 80% power and a 5% false positive rate. Then, we recruited as many participants as possible. A total of 478 customer service employees (71.5% female; $M_{age}$ = 29.4 years, $SD$ = 5.18) from a large company based in China completed the online survey.

All participants first rated their experienced workplace stress based on the item "My stress level right now is _____" (1 = very low, 5 = very high). Next, participants reported their wasting behavior on a five-point scale (i.e., "Sometimes I waste the things I have"; 1 = completely disagree, 5 = completely agree) adapted from Visschers et al. [4]. Then, participants responded to a series of demographic questions.

### 3.1.2. Results

A regression analysis using SPSS 22.0 software revealed that stress was positively associated with wasting behavior: $\beta$ = 0.19, $t$ (476) = 4.10, $p < 0.001$. After including participants' gender (1 = male, 0 = female) and age as covariates, the effect of stress remained significant, $\beta$ = 0.19, $t$ (474) = 4.13, $p < 0.001$.

### 3.2. Study 1

Study 1 was designed to replicate the findings of the pilot study by using samples from an online panel and adopting different types of measurements. Specifically, we predicted that stress would be positively associated with individuals' wasting behavior (Hypothesis 1).

### 3.2.1. Procedure

We sought to recruit as many participants as possible based on the a priori power analysis in the pilot study and actual participant enrollment. Based on these criteria, 300 Chinese participants (45.5% female; $M_{age}$ = 28.3 years, $SD$ = 6.32) participated in our study via Credamo (www.credamo.com, accessed on 23 February 2021). Credamo is a Chinese crowdsourced data collection platform, similar to Amazon Mechanical Turk, that has been widely used in recent studies in psychology and behavioral marketing [62,63].

Participants need to complete measurement of stress and their wasting behavior. Specifically, participants rated their stress level using the adapted stress subscale of Depression Anxiety Stress Scale (DASS; New South Wales, Sydney)) [64]. Participants responded to 14 items describing their experiences in general (e.g., "I find it difficult to relax" and "I find it hard to calm down after something upsets me", 1 = strongly disagree, 7 = strongly agree, $\alpha$ = 0.96; Please see Appendix A, Table A1). Higher scores on this scale indicate more stress.

Next, participants responded to three items regarding their wasting behavior in general, including "I usually discard items I have bought", "I usually leave items unused after I have bought them", and "I usually leave items unused after I have bought them" (1 = strongly disagree, 7 = strongly agree, $\alpha$ = 0.85; Please see Appendix B, Table A2), which were inspired by earlier research on (food) wasting behavior [1,4,60]. Higher scores on this scale reflect greater wasting behavior. Furthermore, participants also completed another scale measuring their food wasting behavior. This scale was adapted from Visschers et al. [4] and includes four items, including "I try to waste no food at all", "I always try to eat all purchased food", "I try to produce only very little food waste", and "I aim to use all leftovers" (1 = strongly disagree, 7 = strongly agree, reverse-coded, $\alpha$ = 0.86; Please see Appendix B, Table A2). We added this scale so as to check the robustness of the effect of stress on wasting behavior. Lastly, the participants answered demographic questions.

### 3.2.2. Results

We first averaged the items measuring participants' general wasting behavior (self-developed), then the items measuring food wasting behavior, and lastly the items related to stress. We used SPSS 22.0 software to process the data. First, participants' scores on the general wasting behavior were positively associated with their scores on food wasting

behavior ($r = 0.40$, $p < 0.001$), which showed that both scales can capture wasting behaviors. A regression analysis revealed that stress was positively associated with general wasting behavior, $\beta = 0.51$, $t\ (298) = 10.17$, $p < 0.001$. The effect of stress on general wasting behavior remained significant after controlling for participants' age and gender (1 = male, 0 = female), $\beta = 0.51$, $t\ (296) = 10.08$, $p < 0.001$. Therefore, these results gave additional support to Hypothesis 1 based on a different set of measures.

As a robustness check, we also tested the effect of stress on individuals' food wasting behavior. A regression analysis revealed that stress was positively associated with food wasting behavior, $\beta = 0.21$, $t\ (298) = 3.71$, $p < 0.001$. The effect of stress on food wasting behavior remained significant after controlling for participants' age and gender (1 = male, 0 = female), $\beta = 0.21$, $t\ (296) = 3.75$, $p < 0.001$. Therefore, we also found the significant association between stress and wasting behavior in a specific domain (i.e., food consumption).

### 3.3. Study 2

Study 2 was guided by two objectives. First, this study was designed to replicate the findings of Study 1 by using student samples and different types of measurements, predicting that stress would be positively associated with individuals' wasting behavior (Hypothesis 1). Second, we examined the mediating role of individuals' self-control. We predicted that high (vs. low) stress would lead to more wasting behavior and that such an effect would be mediated by impaired self-control (Hypothesis 2).

#### 3.3.1. Procedure

As we wished to gather data from college students, we sought to recruit as many participants as possible based on the a priori power analysis in the pilot study and actual student enrollment. Ultimately, 176 college students (75.6% women; $M_{age} = 21.6$ years, $SD = 1.42$) from a university in China participated in our study.

Participants first completed the 14-item perceived stress scale (PSS) adapted from previous literature (e.g., "How often have you been upset because of something that happened unexpectedly", "How often have you felt nervous and 'stressed'"; 1 = never, 7 = very often, $\alpha = 0.84$, adapted from Cohen et al. [65]; Please see Appendix A, Table A1).

Then, participants worked on a 13-item brief self-control scale adapted from prior work [45]. Specifically, they need to indicate how much each of the 13 statements reflect how they typically are (e.g., "I am good at resisting temptation", "People would say that I have iron self-discipline"; 1 = not at all, 7 = very much, $\alpha = 0.74$; Please see Appendix C, Table A3).

Next, participants completed the same three-item scale of general wasting behavior as for Study 1 ($\alpha = 0.78$). After that, as a robustness check, we also asked students to rate their stress level using another scale, adapted from the brief stress subscale of DASS [19]. Participants responded to eight items describing their experiences (e.g., "I find it difficult to relax", "I find it hard to wind down"; 1 = strongly disagree, 7 = strongly agree, $\alpha = 0.92$). Higher scores on this scale indicate higher stress. Finally, the participants answered demographic questions.

#### 3.3.2. Results

SPSS 22.0 software was also used for data processing. We averaged the items related to participants' wasting behavior and then items on the perceived stress scale (PSS). A regression analysis revealed that stress (PSS) was positively associated with wasting behavior, $\beta = 0.27$, $t\ (174) = 3.70$, $p < 0.001$. The effect of stress on wasting behavior remained significant after controlling for participants' age and gender (1 = male, 0 = female), $\beta = 0.20$, $t(172) = 2.65$, $p = 0.009$. Therefore, these results again support Hypothesis 1.

Next, we averaged the scores on items measuring self-control and examined its mediating role in the effect of stress on wasting behavior through path analyses and a bootstrap analysis [66,67]. Path analyses demonstrated that perceived stress was negatively associated with self-control ($\beta = -0.51$, $t\ (174) = -7.81$, $p < 0.001$) and positively associated with

wasting behavior ($\beta = 0.27$, $t$ (174) = 3.70, $p < 0.001$). We then regressed participants' wasting behavior on the stress and self-control. The effect of self-control on wasting behavior was significant ($\beta = -0.26$, $t$ (173) = $-3.13$, $p = 0.002$), whereas the effect of the stress became insignificant ($\beta = 0.14$, $t$ (173) = 1.67, $p = 0.097$). Next, a 5000-sample bootstrap analysis using PROCESS Model 4 [67] again supported this mediation model (indirect effect = 0.24, 95% confidence interval = [0.10, 0.42], excluding zero). Taken together, these findings offered coherent evidence to support Hypothesis 2 by revealing the underlying mechanism of the proposed effect through the mediation model (see Figure 1).

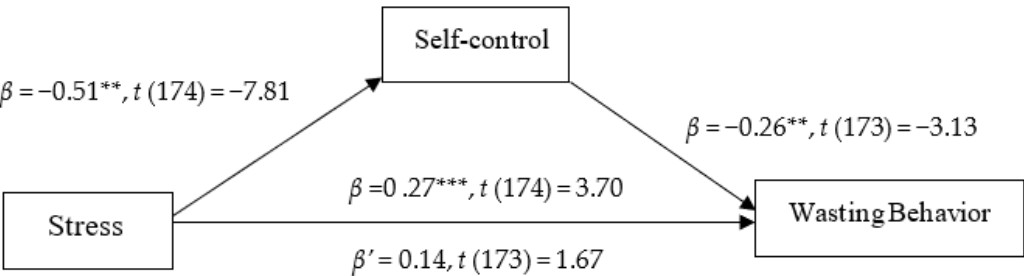

**Figure 1.** Impact of stress on individuals' wasting behavior: mediation model; ** significant at the 0.01 level, *** significant at the 0.001 level.

As a robustness check, we also averaged the items on the brief DASS and performed another bootstrap analysis with the stress measure as the independent variable to test the mediation model. The mediation model still worked (indirect effect = 0.13, 95% confidence interval = [0.04, 0.23], excluding zero), which provides further support for both Hypotheses 1 and 2.

## 4. Conclusions and Discussion

Stress and wasting behavior are two prevalent and important social problems in the modern world. However, researchers have not yet thoroughly investigated the relationship between these issues. The current research filled this knowledge gap by testing how individuals' exposure to stress could influence their wasting behavior. Three studies offered consistent evidence that higher stress leads to more wasting behavior. Additionally, our studies, through a mediation approach, showed that this effect was driven by impaired self-control.

### 4.1. Theoretical and Practical Implications

This research makes several theoretical and practical contributions. First, our work extends the literature testing the influence of stress on human behavior. Specifically, we investigated the relationship between stress and behavior in the consumption domain, which has received little scholarly attention [19]. Additionally, research on similar topics has largely considered humans' purchase and spending decisions regarding the neglect of post-purchase behavior. We bridged this gap by focusing on wasting behavior, which involves the use and disposal of purchased items [1]. Thus, our research can deepen our overall understanding of the behavioral consequences of stress.

Second, our findings enrich the body of knowledge on wasting behavior. This type of behavior is a severe global problem and manifests itself in many aspects [3,16]. However, most relevant research has focused on food waste behavior [16,60]. Drawing on Arkes' [1] theorization, we emphasized "inefficient resource use" as the defining characteristic of wasting behavior and did not limit our investigation to food consumption. For example, both studies 1 and 2 tested participants' general wasting behavior, whereas study 1 also confirmed the positive relationship between general wasting behavior and food wasting behavior. Furthermore, Porpino [16] stated that scarce research has explored the psychological factors (e.g., feelings and emotions) behind wasting behavior. Our research makes

a pioneering attempt to investigate this topic by showing that psychological factors such as stress can lead to wasting behavior.

Third, this research documents the impact of stress on people's wasting behavior through impaired self-control. By using a mediation model, study 2 not only offered evidence to consolidate such a mechanism, but also proposed a feasible way to attenuate the effects of stress on wasting behavior (e.g., increasing self-control abilities). Overall, our findings have meaningful managerial and public policy implications in terms of emotion regulation and waste management. Specifically, previous studies have indicated that people may seek to restore impaired self-control by establishing social connection [68], because close relationships can be energizing and provide resources needed for self-control [69]. Therefore, social interaction, which is interesting, mood enhancing, or self-affirming, is conductive to restore self-control [70,71]. The government could thus start a self-control restoring program to reduce wasting behavior by providing social support, such as organizing community and public benefit activities. In addition, people tend to seek social alternatives if their need for self-control cannot be satisfied in "real" life [72]. For example, they may immerse themselves in a favorite television program [68], build emotional attachment to unfamiliar brands [73], or demonstrate preference for high-effort products (e.g., losing weight and maintaining a clean home) [74]. Thus, marketers could consider maintaining consumers' sense of self-control through offering products and services with relevant psychological functions.

What is more, our research provides a fresh perspective to explain how individuals behave in a state of feeling stressful. The literature has shown that purchase or consumption activities can help people cope with stress. For instance, Chen et al. [75] observed that individuals were more willing to acquire utilitarian products to alleviate their feeling of stress because consumers believed these purchased items could provide enduring utility. Similarly, Durante and Laran [19] also found stress would motivate people to spend more on necessities. However, our findings implied that when regulating purchased items, people with higher stress failed to use these items efficiently, leading to wasted utility. These findings unveil an intriguing paradox; that is, stress might lead to a mismatch between people's expected and actual outcomes of certain behaviors, such as consumption.

*4.2. Future Research*

This paper presents several directions for future work. First of all, our research marks an early attempt to explore the role of psychological states in individuals' wasting behavior. Future studies can examine other psychological states, such as (1) specific emotions (e.g., shame and guilt), (2) feelings aroused by environmental factors (e.g., a sense of confinement), and (3) one's sense of self-identity (e.g., self-concept clarity). Consideration of these psychological antecedents can enrich our understanding of why people waste the things they have, even though individuals are aware of and averse to being wasteful in most cases [1,3].

Second, the current research takes an initial step to adopt survey methods to evaluate general wasting behavior. Future work can potentially (1) develop other instruments to capture general wasting behavior and (2) use big data and field experiments to study this topic. These efforts can lay a solid methodological foundation for research in this field. In addition, applying multiple approaches could allow us to consider more factors that may jointly influence wasting behavior. For example, Fami et al. [76] demonstrated a comprehensive behavioral model to explain household food wasting behavior from the perspective of motivation and ability. It revealed a variety of antecedents associated with the rise of food waste, such as information use, knowledge level, age, and household size. Future research can combine these individual features with a self-control mechanism to manage wasting behavior. Specifically, for instance, the more knowledge consumers have, the more capable they are of controlling themselves under stress, which may in turn reduce waste.

Third, our research proposes that impaired self-control drives wasting behavior. Scholars can extend our work by identifying ways to facilitate waste management psychologically. For example, according to our theorization, better waste management is associated with careful planning, self-monitoring, and execution, which are three fundamental components of self-control. Therefore, future research can explore how to reduce wasting behavior through changes in these aspects.

**Author Contributions:** Conceptualization, K.Z. and Y.C.; Data curation, Y.C.; Funding acquisition, K.Z. and Y.C.; Methodology, K.Z. and Y.C.; Writing—original draft, K.Z.; Writing—review & editing, Y.C. All authors have read and agreed to the published version of the manuscript.

**Funding:** This work was supported by the National Natural Science Foundation of China [grant number: 72172087] and Chenguang Program [grant number: 18CG45] awarded to the first author; the National Natural Science Foundation of China [grant number: 71602111] awarded to the second author.

**Institutional Review Board Statement:** The research was conducted in accordance with the Declaration of Helsinki, and approved by the Institutional Review Board of the SILC Business School at Shanghai University (protocol code: Not applicable, 1 February 2020).

**Informed Consent Statement:** Informed consent was obtained from all subjects involved in the study.

**Data Availability Statement:** The data presented in this research are available on request from the first author and the corresponding author. The data are not publicly available to protect the confidentiality of the participants.

**Conflicts of Interest:** The authors declare no conflict of interest.

## Appendix A

**Table A1.** Stress scales in studies.

| Scale | Items |
|---|---|
| Depression Anxiety Stress Scale (DASS), stress subscale (Study 1, Study2) | * 1. I find it difficult to relax<br>2. I find it hard to calm down after something upsets me<br>* 3. I find it hard to wind down<br>* 4. I feel that I am using a lot of nervous energy<br>* 5. I am in a state of nervous tension<br>* 6. I find myself getting upset rather easily<br>* 7. I find myself getting agitated<br>8. I find myself getting upset by quite trivial things |
|  | 9. I tend to over-react to situations<br>* 10. I find that I am very irritable<br>* 11. I feel that I am rather touchy<br>12. I am intolerant of anything that keeps me from getting on with what I was doing<br>13. I find myself getting impatient when I am delayed in any way (e.g., lifts, traffic lights, and being kept waiting)<br>14. I find it difficult to tolerate interruptions to what I am doing |
| Perceived Stress Scale (PSS) (Study 2) | 1. How often have you been upset because of something that happened unexpectedly<br>2. How often have you felt that you were unable to control the important things in your life<br>3. How often have you felt nervous and "stressed" |
|  | [a] 4. How often have you dealt successfully with irritating life hassles<br>[a] 5. How often have you felt that you were effectively coping with important changes that were occurring in your life<br>[a] 6. How often have you felt confident about your ability to handle your personal problems<br>[a] 7. How often have you felt that things were going your way<br>8. How often have you found that you could not cope with all the things that you had to do<br>[a] 9. How often have you been able to control irritations in your life<br>[a] 10. How often have you felt that you were on top of things<br>11. How often have you been angered because of things that happened that were outside of your control<br>12. How often have you found yourself thinking about things that you have to accomplish<br>[a] 13. How often have you been able to control the way you spend your time<br>14. How often have you felt difficulties were piling up so high that you could not overcome them |

Note: * Items included in the brief edition; [a] Reverse-coded items.

## Appendix B

**Table A2.** Measures of wasting behavior in studies.

| Scale | Items |
|---|---|
| General wasting behavior (Study 1 and Study 2) Food wasting behavior (Study 1) | 1. Usually, I cannot eat up the food I have bought. 2. I usually discard items I have bought. 3. I usually leave items unused after I have bought them. [a] 1. I try to waste no food at all [a] 2. I always try to eat all purchased food [a] 3. I try to produce only very little food waste |

Note: [a] Reverse-coded items.

## Appendix C

**Table A3.** Self-control scale in study 2.

| Scale | Items |
|---|---|
| Self-control (Study 2) | 1. I am good at resisting temptation [a] 2. I have a hard time breaking bad habits [a] 3. I am lazy [a] 4. I say inappropriate things [a] 5. I do certain things that are bad for me, if they are fun 6. I refuse things that are bad for me [a] 7. I wish I had more self-discipline 8. People would say that I have iron self-discipline |
|  | [a] 9. Pleasure and fun sometimes keep me from getting work done [a] 10. I have trouble concentrating 11. I am able to work effectively toward long-term goals [a] 12. Sometimes I can't stop myself from doing something, even if I know it is wrong |

Note: [a] Reverse-coded items.

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
