# Peer review of "The Effect of Stress on Individuals’ Wasting Behavior: The Mediating Role of Impaired Self-Control"

_sustainability, doi:10.3390/su14031176_

Round 1

Reviewer 1 Report

Review comments

  1. The first paragraph of the introduction explained the definition and two manifestations of wasting behavior, while the second paragraph mainly elaborated on the measurement and evaluation of wasting behavior, both of which were brief introductions to wasting behavior. Would it not be better to combine these two paragraphs for the reader’s coherently?
  2. The first sentence in the third paragraph of the introduction stated that a large amount of resources, such as food and water, were wasted each It is recommended that a specific number could be added here to see the waste of resources more intuitively, and also to further highlight the urgency of exploring the causes of wasting behavior given the scarcity of resources.
  3. In section 1.1 (Research about Stress), the definition of stress, the antecedents of stress (stress sources) and the consequences of stress were discussed. In lines 89 to 92, it was written that stress affected the social behaviors and health-related behaviors of individuals, where these studies related to stress and individual behaviors could be analyzed and elaborated in depth, so that readers gain deeper understanding of the relationship between stress and individual behaviors.
  4. In 1.1 section (Research about Stress), the emphasis should be on strengthening the in-depth elaboration of the relationship between stress and personal behavior, rather than mainly on the depiction of stress itself
  5. 1A and 1B are both designed to test the impact of stress on individual wasting behavior. 1A chooses company employees as the survey subjects, and 1B chooses students as the survey subjects. Is this survey subject selected randomly or is there any consideration?
  6. In the methods section, what software was used to data process of the article?
  7. This article investigated the mechanisms of stress on individual wasting behavior by using a mediation model with impaired self-control as a mediation variable. In the discussion section, the article mentioned that this study has important management and public policy significance, but the specific significance was not written out, suggesting that the policy significance of this study could be elaborated in detail.

Reviewer 2 Report

The article concerns the interesting issue of determinants of various aspects of behavior resulting in wasting (goods, time, money ...). Among the determinants of these behaviors, attention was paid to self-control and the level of stress. The authors refer to, inter alia, to selected research on consumerism, stress and self-control and on this basis they formulate two hypotheses and a model of their own research.

The tested compound model is clear. The expected relationships were confirmed in the statistical analysis.

The article consists of two studies. The first study involved two different groups of subjects (1A and 1B).

The first one consisted of employees from one of the companies in China. One item (only one) was used to measure stress at work and a 5-point scale (adapted from Visschers et al., 2016) was used to measure wasteful behavior. The age of the respondents in the first study was not given - based on  “Mean”, it can be assumed that the study participants were in early adulthood.

The second group (1B) also consisted of working people - although one of the statements suggests that perhaps there were also students? (page 4). This time a different tool was used in measuring stress than in the first group (1A). Namely, 14 items from the subscale of Depression Anxiety Stress Scale (DASS; Lovibond & Lovibond, 1995) were used. Three items were used to measure wasting behavior (in general) - constructed on the basis of the texts: Arkes, 1996; Visschers et al., 2016; van der Werf et al., 2020 and the second tool, which consisted of four items from the Visschers et al. scale. (2016).

These studies verified hypothesis 1.

The second study involved college students in early adulthood (or in transition from adolescence to early adulthood - depending on the classification adopted) - I conclude from the “mean” age.

A self-control measurement was performed (using the 13-item brief self-control scale, Tangney et al., 2004). The level of stress was measured using two tools: (1) 14-item perceived stress scale (PSS) Cohen, 1983 ), and (2) 8 items taken from the brief stress subscale of DASS (Durante & Laran, 2016). General wasting behavior was also measured using 3-item - as Study 1B.

The results of the statistical analysis confirmed hypothesis 1 and hypothesis 2.

It is not obvious whether the tools used have had an adaptation to the population in China? It would be good to clarify this point.

The article reads well, the language and style are reader-friendly. I do not understand why it consists of two studies: 1 (A + B) and 2 - because the research results in study 2 allowed for the verification of both hypotheses.

I am somewhat unsatisfied with the last part of the article. On the basis of the introduction, I expected the authors to point out the usefulness of their research results for building programs enhancing the level of self-control.

I do not see in the text the basis for making a reference to, for example, waste management in the last part of the article.

I expected an in-depth reflection on the relationship of self-control, stress and wasteful behavior - even in the context of the Chinese population (fact - very diverse). The results obtained by the authors indicate the mediating role of self-control between stress and wasting behavior. However, from a broader perspective - outlined in the first part of the article, e.g. by referring to the Covid-19 pandemic - it would be justified to repeat the research (Study 2) in a different context. Can it be ruled out that a lower / low level of self-control is a specific way to reduce stress? - - allows some people to distract attention/thinking from threats?

Reviewer 3 Report

This is an interesting study that seeks the role a psychological state, stress, plays in affecting people’s wasting behaviour. My major comments are as follows:

  1. Reorganise the structure.

Try to separate Introduction with Literature review. Section “1.1. Research about Stress,” and “1.2. The Effect of Stress on Wasting Behavior: The Mediating Role of Impaired Self-Control” are apparently not subsections of Introduction, instead, they are literature reviews that support your current study. Accordingly, section “1.3. Overview of Studies” is your research design that can be situated in the Methodology part. At the end of your introduction, you may briefly introduce the overall frame of this article. Several added sentences summarising the experiments part could be good.

The paper could (but Not necessarily) follow the structure as listed:

Introduction – Literature review – Methodology – Results – Conclusion

  1. Do you use any existing model (e.g., behaviour models) to test your hypothesis?
  2. In terms of the research design, study 1A should be the pilot test of your primary study, which is study 1B. Study 1A seems to be redundant. Meanwhile, three studies are conducted in different scenarios with different types of participants, especially Study 2, the participants are all colleague students, which definitely not able to represent other age groups. Did you use other data analysis methods besides regression analysis and path analysis? And which software did you use?
  3. Try to separate results with your methodology. Now they are mixed, which makes the paper hard to comprehend.
  4. It would be better if you can highlight your own limitations.

Minor comments:

  1. Line 89 – 96 stated the research gap that little attention has been paid to “the impact of stress on consumers’ everyday behavior.” The author could write more about the importance of studying consumers’ daily behaviours.
  2. Line 104, the format of the words “Bernheim et al. (2015)” need to be adjusted.

As the author cited, “stress represented a major mental health concern for people worldwide.” Understanding the impact of which is important. I am looking forward to reading the revised version of this paper.

Round 2

Reviewer 2 Report

The second version of the article is better than the first version. A pilot study has been introduced - instead of Study 1A.
However, regarding the pilot study, the question arises: on what basis did the authors assume that the stress level at the time of the study was the same as that of the waste situation to which the questions referred? (Authors: "we measured stress experienced by employees at their workplace and predicted that it would be positively correlated with self-reported wasting behavior".)
In the second study, other measures of stress were used to measure "overall" stress ("stress level using the adapted stress subscale of Depression Anxiety Stress Scale (DASS; Lovibond & Lovibond, 1995)".
He suggests that the authors calculate the Cronbach's alpha reliability index for the tools used and their research (data).

I wonder how the government could start a plan to restore self-control - maybe it would be better to write that it is about government programs

It is necessary to read the text carefully to remove the "remnants" of the earlier version, eg "Study 1 was designed to replicate the findings of Study 1A by using s ...", this also applies to the abstract. There are some minor linguistic mistakes.

The second version of the article is better than the first version. A pilot study has been introduced - instead of Study 1A.
However, regarding the pilot study, the question arises: on what basis did the authors assume that the stress level at the time of the study was the same as that of the waste situation to which the questions referred? (Authors: "we measured stress experienced by employees at their workplace and predicted that it would be positively correlated with self-reported wasting behavior".)
In the second study, other measures of stress were used to measure "overall" stress ("stress level using the adapted stress subscale of Depression Anxiety Stress Scale (DASS; Lovibond & Lovibond, 1995)".
He suggests that the authors calculate the Cronbach's alpha reliability index for the tools used and their research (data).

I wonder how the government could start a plan to restore self-control - maybe it would be better to write that it is about government programs

It is necessary to read the text carefully to remove the "remnants" of the earlier version, eg "Study 1 was designed to replicate the findings of Study 1A by using s ...", this also applies to the abstract. There are some minor linguistic mistakes.

The second version of the article is better than the first version. A pilot study has been introduced - instead of Study 1A.
However, regarding the pilot study, the question arises: on what basis did the authors assume that the stress level at the time of the study was the same as that of the waste situation to which the questions referred? (Authors: "we measured stress experienced by employees at their workplace and predicted that it would be positively correlated with self-reported wasting behavior".)
In the second study, other measures of stress were used to measure "overall" stress ("stress level using the adapted stress subscale of Depression Anxiety Stress Scale (DASS; Lovibond & Lovibond, 1995)".
He suggests that the authors calculate the Cronbach's alpha reliability index for the tools used and their research (data).

I wonder how the government could start a plan to restore self-control - maybe it would be better to write that it is about government programs

It is necessary to read the text carefully to remove the "remnants" of the earlier version, eg "Study 1 was designed to replicate the findings of Study 1A by using s ...", this also applies to the abstract. There are some minor linguistic mistakes.

The second version of the article is better than the first version. A pilot study has been introduced - instead of Study 1A.
However, regarding the pilot study, the question arises: on what basis did the authors assume that the stress level at the time of the study was the same as that of the waste situation to which the questions referred? (Authors: "we measured stress experienced by employees at their workplace and predicted that it would be positively correlated with self-reported wasting behavior".)
In the second study, other measures of stress were used to measure "overall" stress ("stress level using the adapted stress subscale of Depression Anxiety Stress Scale (DASS; Lovibond & Lovibond, 1995)".
He suggests that the authors calculate the Cronbach's alpha reliability index for the tools used and their research (data).

I wonder how the government could start a plan to restore self-control - maybe it would be better to write that it is about government programs

It is necessary to read the text carefully to remove the "remnants" of the earlier version, eg "Study 1 was designed to replicate the findings of Study 1A by using s ...", this also applies to the abstract. There are some minor linguistic mistakes.

The second version of the article is better than the first version. A pilot study has been introduced - instead of Study 1A.
However, regarding the pilot study, the question arises: on what basis did the authors assume that the stress level at the time of the study was the same as that of the waste situation to which the questions referred? (Authors: "we measured stress experienced by employees at their workplace and predicted that it would be positively correlated with self-reported wasting behavior".)
In the second study, other measures of stress were used to measure "overall" stress ("stress level using the adapted stress subscale of Depression Anxiety Stress Scale (DASS; Lovibond & Lovibond, 1995)".
He suggests that the authors calculate the Cronbach's alpha reliability index for the tools used and their research (data).

I wonder how the government could start a plan to restore self-control - maybe it would be better to write that it is about government programs

It is necessary to read the text carefully to remove the "remnants" of the earlier version, eg "Study 1 was designed to replicate the findings of Study 1A by using s ...", this also applies to the abstract. There are some minor linguistic mistakes.

The second version of the article is better than the first version. A pilot study has been introduced - instead of Study 1A.
However, regarding the pilot study, the question arises: on what basis did the authors assume that the stress level at the time of the study was the same as that of the waste situation to which the questions referred? (Authors: "we measured stress experienced by employees at their workplace and predicted that it would be positively correlated with self-reported wasting behavior".)
In the second study, other measures of stress were used to measure "overall" stress ("stress level using the adapted stress subscale of Depression Anxiety Stress Scale (DASS; Lovibond & Lovibond, 1995)".
He suggests that the authors calculate the Cronbach's alpha reliability index for the tools used and their research (data).

I wonder how the government could start a plan to restore self-control - maybe it would be better to write that it is about government programs

It is necessary to read the text carefully to remove the "remnants" of the earlier version, eg "Study 1 was designed to replicate the findings of Study 1A by using s ...", this also applies to the abstract. There are some minor linguistic mistakes.

Reviewer 3 Report

Thank you very much for your improvement. However, the paper still needs some revision before it can be considered by the journal.

  1. A short paragraph stating your overall research design at the beginning of your methodology section is highly recommended.
  2. It is very important to seperate your research results and discussion to avoid confusing your readers. 
  3. There exists ample research concerning behaviour models (e.g.,Fami, H. S., Aramyan, L. H., Sijtsema, S. J., & Alambaigi, A. (2019)) Currently the model looks quite simple.
